# MULTI-EXPERT COLLABORATION: ENHANCING HETEROGENEOUS KNOWLEDGE INDEPENDENCE AND ALIGNMENT IN KNOWLEDGE DISTILLATION

## ABSTRACT

Heterogeneous multi-teacher Knowledge distillation attempt to learn a versatile student neural network from multiple pre-trained heterogeneous teachers. But current methods face issues with a lack of independence and alignment in heterogeneous knowledge. To address this issue, we propose a novel method called Multi-Expert Collaboration (MEC). Our approach aggregates multiple expert classifiers within the student model, replacing the conventional single-head architecture. By ensuring that each expert's independent classifier operates without interfering with others, we enhance the independence of heterogeneous knowledge. Inspired by Helmholtz Free Energy (HFE) theory, we introduce an anchor-based HFE self-normalization strategy to align the heterogeneous knowledge effectively. This method ensures consistent energy levels across all classifiers, allowing the appropriate classifier to achieve the highest confidence for in-distribution data. Extensive experiments on CIFAR-100 and ImageNet-100 datasets demonstrate that MEC significantly outperforms existing heterogeneous multi-teacher knowledge distillation methods, achieving an average accuracy improvement of over 10%.

## 1 INTRODUCTION

Deep neural networks have achieved remarkable success across various applications, and numerous deep network models optimized for different tasks and trained on diverse datasets have been made publicly available, providing researchers with a rich repository of model resources. Despite this abundance, effectively leveraging these heterogeneous teacher networks to train a student model capable of handling multiple tasks remains a pressing and unresolved challenge in the field.

Knowledge Distillation (KD) Hinton et al. (2015) methods are primarily used to transfer knowledge from a single complex teacher model to a light-weight student model. The core idea is to transfer knowledge by having the student model mimic the output logits or soft targets generated by the teacher model, thereby capturing the teacher's learned representations and generalization capabilitiesBeyer et al. (2022); Gong et al. (2023); Agand (2024). However, heterogeneous multi-teacher models involve multiple teachers trained based on different architectures, training data, and task objectives. KD methods, being limited to homogeneous knowledge transfer, are unable to effectively merge knowledge from multiple pre-trained heterogeneous teacher models.

Recently, researchers have explored a heterogeneous multi-teacher knowledge distillation approach-Knowledge Amalgamation (KA) Shen (2019); Thadajarassiri et al. (2021); Xu et al. (2022); Zhang et al. (2023); Gao et al. (2024), which aggregates the knowledge of multiple teachers into a student model. As shown in Figure 1a, KA methods usually adopt a dual approach of aligning common features and extracting soft targets.

However, current heterogeneous multi-teacher knowledge distillation only refines the student's knowledge by simply concatenating logits of teacher models and select the class with the highest in the student's logit. As the number of teachers increases, the mutual interference between heterogeneous knowledge in the student model will affect the performance of the model. When multiple expert models' predictions are simply integrated into a student model, the heterogeneity of knowledge makes it difficult for the student model to determine which knowledge is more representative or accurate.

To achieve independence and alignment of heterogeneous knowledge, this paper proposes a novel method called Multi-Expert Collaboration (MEC), aiming to utilize these pre-trained networks specialized in different tasks (expert networks) to learn a multi-skilled student network. As shown in Figure 1b, our method first aggregates multiple expert classifiers within the student model instead of using a single-head approach. By ensuring that each expert's independent classifier does not interfere with others, we enhance the independence of heterogeneous knowledge. Inspired by studies on Helmholtz Free Energy (HFE) Liu et al. (2020), we observe that for a given classifier, in-stage data typically exhibit higher free energy (i.e., higher confidence scores) compared to out-stage data. To tackle the challenge of aligning heterogeneous knowledge, we adopt an anchor-based HFE self-normalization strategy to ensure that all classifiers operate at a consistent energy level. This method effectively aligns the heterogeneous knowledge from different experts, ensuring that the appropriate classifier achieves the highest confidence for in-stage data, thereby enhancing the overall alignment of heterogeneous knowledge in the student model.In summary, our contributions are three-fold:

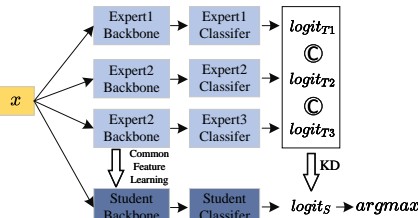

(a) In KA method: the logit of x is evaluated under each expert model, knowledge distillation on the logit of student models through simple concatenation, and the class with the highest one is chosen in the student logit.

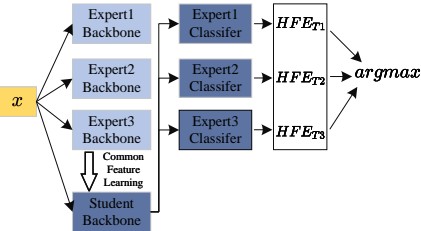

(b) In MEC method: the HFE of x is evaluated under each expert classifer, HFE self normalizes in energy space, and the class with the highest one is chosen.

Figure 1: Comparison of two paradigms. The parameters of the light blue module are frozen, while the parameters of the dark blue module can be learned.

- We propose a multi expert collaborative method to address the independence and alignment issues of heterogeneous knowledge in heterogeneous multi teacher knowledge distillation.

- Multi-expert representation learning is employed to obtain a universal feature extraction backbone, while the multi-head classifier ensures the independence of heterogeneous knowledge. The anchor-based HFE self-normalization method further ensures the alignment of this heterogeneous knowledge.

- Extensive experiments on two benchmark datasets demonstrate the superior performance of the proposed multi-expert collaboration paradigm, achieving an average accuracy improvement of over 10% compared to other heterogeneous multi-teacher knowledge distillation methods

## 2 RELATED WORK

### 2.1 KNOWLEDGE DISTILLATION

Knowledge Distillation (KD) Hinton et al. (2015) is an efficient technique for model reuse and compression, where a smaller student model is trained to replicate the behavior of a larger, more complex teacher model. The core idea is to transfer knowledge by having the student model mimic the output logits or soft targets generated by the teacher model, thereby capturing the teacher's learned representations and generalization capabilities. The goal is to minimize the difference in probabilistic outputs between the teacher and the student models. Research in this area primarily focuses on exploring the potential of knowledge transfer between the teacher and student models. Various approaches have been investigated, including aligning intermediate layersBeyer et al. (2022); Gong et al. (2023), utilizing auxiliary teacher modelsMirzadeh et al. (2020), and selecting expert modelsAgand (2024).

### 2.2 KNOWLEDGE AMALGAMATION

Knowledge amalgamation, as an extension of Knowledge distillation, Shen (2019) was first introduced to merge knowledge from multiple expert models into a single student model suitable

for all tasks, particularly in classification tasks. Traditionally, KA methods Ye et al. (2019); Shen et al. (2019); Luo et al. (2019); Xu et al. (2022) train the student model by mimicking the outputs of teacher models-(classification score learning) -and/or their intermediate representations (feature learning). The scenario of KA has gained attention due to the prevalence of publicly available pre-trained models with different architectures. In such cases, the student model cannot directly learn from the features of each block of the teacher models as in typical homogeneous settings. To balance this heterogeneity, most previous KA methods—such as data-free knowledge amalgamation Ye et al. (2020), semi-supervised knowledge amalgamation Thadajarassiri et al. (2021), weighted amalgamation strategies Luo et al. (2020), and model-heterogeneous aggregated training Xu et al. (2022); Zhang et al. (2023); You et al. (2024); Gao et al. (2024) —mainly rely on classification score learning to achieve heterogeneous multi-teacher knowledge distillation.

## 3 PROBLEM FORMULATION

Multi-expert collaboration is defined as follows. Assume that given $E = \{E_1, E_2, ..., E_N\}$ representing $N$ experts, for an expert $A_n$, its data is represented as $D_n = \{x_i, y_i\}_{i=1}^{M_n}$, where $M_n$ represents the total number of samples for $E_n$, $x_i$ denotes a sample and $y_i$ denotes the corresponding label. The goal of multi-expert collaboration is to derive a generalist that classifies all classes as $D = \bigcup_{i=1}^{N} D_i$. To be specific, for any two experts $E_i$ and $E_j$ where $i \neq j$, $D_i \neq D_j$, which means that each expert classifies independent and different tasks, and the data class sets they possess are unique. After collaboration, generalist can infer the classes $Y_{E_i} \cup Y_{E_j}$, where $Y$ is a label space.

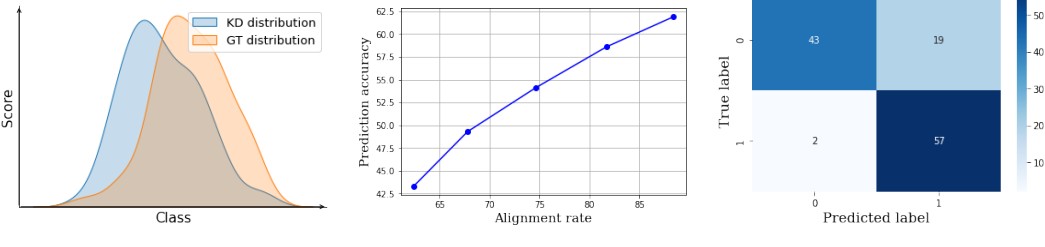

(a) Heterogeneous knowledge alignment issue.

(b) Correlation between alignment rate and prediction Accuracy.

(c) misaligned heterogeneous knowledge distillation analysis.

Figure 2: Analysis of heterogeneous knowledge alignment in heterogeneous multi-teacher knowledge distillation.

**Heterogeneous knowledge alignment**. each expert focuses on a specific task without being exposed to others, which may lead to overly confident predictions that could mislead the student model. As shown in Figure 2a, KD logit distribution is not aligned with GT logit distribution. Using misaligned logit for knowledge distillation can negatively impact the student's learning process. Through gradient analysis, $z_s^{(i)}$ is student logit,

$$\mathcal{L} = (1 - \alpha)\mathcal{L}_{CE} + \alpha\mathcal{L}_{KD}$$
$$= (1 - \alpha)(-\log p_s^{(i)}) + \alpha(-\sum_{i=1}^{C} p_t^{(i)} \log p_s^{(i)}) \quad (1)$$

$$\frac{\partial \mathcal{L}}{\partial z_s^{(i)}} = (1 - \alpha)(p_s^{(i)} - y^{(i)}) - \alpha(p_s^{(i)} - p_t^{(i)}) \quad (2)$$

$y^{(i)} = 0$, but $p_t^{(i)} \approx 1$, the gradient of KL provides incorrect guidance to $p_s^{(i)}$, causing the student model to lean towards the wrong class $i$.

We analyzed the relationship between the alignment rate of the KD distribution with the GT distribution and the prediction accuracy of the student model during the knowledge distillation process on the CIFAR100 dataset. As shown in Figure 2b, there is a positive correlation between prediction accuracy and the alignment rate. Further analysis of the prediction results is presented in Figure

2c. During the distillation training process, the alignment rate for class 0 was 62.4%, with 26.1% of the samples having logits corresponding to class 1. In the testing phase of the student model, the prediction accuracy for class 0 was 43%, with 19% of the samples being incorrectly predicted as class 1. This demonstrates that maintaining a high alignment rate in the heterogeneous knowledge alignment process is crucial for achieving better student model performance.

To achieve heterogeneous knowledge alignment, the output of the expert classifiers should meet the following criteria: Criterion 1: Each stage classifer should have a higher output confidence score for the data within the stage it belongs to (i.e., in-stage data) than others (i.e., out-stage data);Criterion 2: The confidence scores for in-stage data should be consistent across all stages.These criteria ensure that the knowledge from heterogeneous experts is effectively integrated, providing a more reliable foundation for the student model's learning process.

**Heterogeneous knowledge independence**. In heterogeneous multi-teacher knowledge distillation, the student model's output layer is a fully connected layer used to classify all classes, requiring model parameters to be shared among all classes. As shown in Figure 3, when the number of experts increases, the integration of diverse and potentially conflicting knowledge from different experts leads to interference among the heterogeneous knowledge domains, resulting in decreased classification accuracy. Each expert model may have been trained on different datasets with unique class distributions and feature representations, leading to distinct decision boundaries. When these heterogeneous knowledge representations are combined within a single-head classifier, the conflicting features and decision boundaries can interfere with each other. This interference makes it challenging for the shared parameter matrix $W \in \mathbb{R}^{C \times D}$ ($D$ is the feature dimension) to effectively represent each class accurately. The overlapping and inconsistent information from various experts complicates the learning process, causing the model to misclassify inputs or fail to capture the essential characteristics of each class. Consequently, the interference among heterogeneous knowledge hinders the model's ability to generalize across tasks, ultimately leading to degraded classification performance.

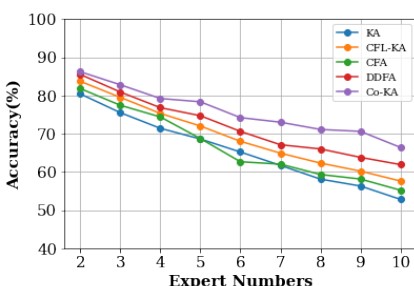

Figure 3: Poor classifier scalability.

# 4 METHOD

In this section, we primarily introduce the paradigm of multi-expert collaboration. Through collaborative efforts with multiple experts, generalist can handle all previous tasks, using only a small amount of data samples for training.

## 4.1 MULTI-EXPERT COLLABORATION FRAMEWORK

We conducted an in-depth investigation on leveraging multiple experts to achieve comprehensive learning, enabling a vanilla to evolve into a generalist. We designed a unified multi-expert collaboration framework U($\cdot$), which allows for moderate information exchange between multiple experts, in order to take advantage of the complementarity of each expert and achieve generalization.

$$SA = \varphi(\bigcup_{i=1}^{N} E_i), \tag{3}$$

where $SA$ is the generalist, $E_i$ denotes an expert, $\varphi(\cdot)$ denotes the approach of collaboration.

As depicted in Figure 4, we decouple the feature representation adaptation from the final classifier in the deep network. This decoupling is facilitated through two primary modules: Multi-Expert Representation Learning and Classifier Adaptation Learning, which together enable effective multi-expert collaboration. For clarity, we will provide a detailed explanation of each module within the multi-expert collaboration framework.

Figure 4: Illustration of multi-expert collaboration. Model is decoupled into feature representation and classifier. Multi-expert representation learning aims to create a universal feature representation. Classification adaptation learning aggregates experts' classifier into generalized classifiers, ensuring the independence of experts classifiers and minimizing interference to the greatest extent possible.

### 4.1.1 MULTI-EXPERT REPRESENTATION LEARNING

We employ multi-expert representation learning to construct a universal feature extraction backbone. This involves transforming both the experts' features and the generalist's features into a common feature space. We then minimize two loss terms: feature loss $L_M$, which encourages generalist's feature to align with expert's feature in common feature space, and reconstruction loss $L_R$, which ensures that transformed expert's feature can be remapped to the original space with minimal error.

To align the output feature dimensions of the experts and generalist, we apply a $1 \times 1$ convolutional kernel, which standardizes the output length regardless of input sizes. Directly averaging the original features $F_G$ (from the generalist) and $F_i$ (from expert $E_i$) can introduce feature heterogeneity. To address this, we introduce a small network shared by both the expert and generalist models. This network converts $F_i$ and $F_G$ into a common feature space, resulting in $f_i$ and $f_G$, which are half the size of the original features. These transformed features are then projected into a low-dimensional subspace, where we constrain their variations to effectively distill the important features.

To amalgamate knowledge from heterogeneous experts, we build a domain-invariant feature space between generalist and experts via KL scatter, computed as follows:

$$D_{KL_i}(f_G \| f_i) = H(f_G, f_i) - H(f_G), \tag{4}$$

where $H(f_G, f_i)$ denotes the cross-entropy of $f_i$ and $f_G$, and $H(f_G)$ denotes the entropy of $f_G$.

The pairwise KL loss between each expert and generalist is expressed as the overall difference in shared space.

$$\mathcal{L}_M = \sum_{i=1}^{N} D_{KL_i}, \tag{5}$$

To further enhance joint representation learning, we add a reconstruction loss between the feature spaces of the original experts. Let $F_i'$ denote the reconstructed features of the expert model and the reconstruction loss is defined as:

$$\mathcal{L}_R = \sum_{i=1}^{N} \|F_i' - F_i\|_2, \tag{6}$$

### 4.1.2 HETEROGENEOUS KNOWLEDGE INDEPENDENCE AND ALIGNMENT

Inspired by studies on Helmholtz free energy (HFE) Liu et al. (2020); Wang et al. (2023), we observed that for a given classifier, in-stage data generally exhibits higher free energy (i.e., higher confidence scores) compared to out-stage data. To tackle the challenge of heterogeneous knowledge alignment, we employ an anchor-based HFE self-normalization strategy, ensuring that all classifiers operate at consistent energy levels. This approach effectively aligns the knowledge across different experts, ensuring that the appropriate classifier achieves the highest confidence score for in-stage data, thereby enhancing the overall alignment and integration of heterogeneous knowledge within the generalist model.

Specifically, we employ a effective energy self-normalization loss $\mathcal{L}_{al}$ to mitigate the conflict between $\mathcal{L}_{ce}$ and $\mathcal{L}_{kd}$. The loss $\mathcal{L}_{kd}$ constrains the free energy of each classifier to a fixed anchor $\Delta$ for improved knowledge interaction, which we detail further.

The generalist consists of a backbone $f_b^s(\cdot)$ and a classifier $f_c^s(\cdot)$. evolving from vanilla to generalist to classify all classes. The generalist classifier includes the same number of classification heads as the experts, initialized with the weights from the expert heads. For each classification head, represented by $f_c^{t_i}(\cdot)$, the expert transfers features extracted by $f_b^s(\cdot)$ to obtain the corresponding HFEs. The classifier with the highest HFE is used for predictions. We then define a normal energy function for a given input-label pair $(x, y)$.

$$E^m(x, y) = -h^m(x)[y], \tag{7}$$

where $h^m(x) = f_c^{t_m}(f_b^s(x))$ is the logit of the $m$-th expert's classifier, and $h^m(x)[y]$ is the logit value of $y \in Y_m$.

Then the Helmholtz free energy can be expressed as the negative log partition function:

$$\mathcal{F}^m(x) = -log \sum_{y \in Y_m} exp(-E^m(x, y)), \tag{8}$$

To align the HFE of different expert classifiers in the same space, we employ $\mathcal{L}_{al}$, which constrains the HFE of each classifier with a fixed anchor $\Delta$, as Eq. 7.

$$\mathcal{L}_{al}^m = \mathbb{E}_{x \sim D_m}(\mathcal{F}^m(x) - \Delta)^2, \tag{9}$$

Assuming $y^{t_i}$ represents the prediction of expert, $\hat{y}^s$ represents the prediction of the generalist, and its KL scatter is denoted as :

$$\mathcal{L}_{kd}\left(\hat{p}^{s_i} \,||\, p^{t_i}\right) = \mathbb{E}_{x \sim D}[log \frac{\hat{p}^{s_i}/T}{p^{t_i}/T}], \tag{10}$$

where $T$ is the temperature parameter. To train the entire network, the overall loss consists of three parts:

$$\mathcal{L} = \mathcal{L}_{ce}(p^s, y) + \lambda_1(\mathcal{L}_M + \mathcal{L}_R) + \lambda_2(\mathcal{L}_{al} + \sum_i^N \mathcal{L}_{kd}), \tag{11}$$

where $\mathcal{L}_{ce}(p^s, y)$ is the cross-entropy loss, $\lambda_1$ and $\lambda_2$ are the trade-off parameters.

In the inference stage, we input the features into a classifier containing 5 classification heads, the final prediction is made by obtaining the classifier with the highest HFE, as follows:

$$m^* = argmax(-\mathcal{F}^m(x)), \tag{12}$$

Then, the final prediction can be obtained as:

$$p^s = f_c^{m^*}(f_b^s(x)) \tag{13}$$

## 5 EXPERIMENTS

We performed a comparative evaluation of various baselines on two datasets on which our method achieved consistently better or equivalent performance. In the following sections, we provide details of the datasets, baselines, experimental setup, quantitative results, and analysis.

## 5.1 EXPERIMENTS SETUP

**Datasets** We validate our method on the widely used benchmark **CIFAR-100** Krizhevsky (2009) and **ImageNet-100** Deng et al. (2009). CIFAR-100 is a classification dataset with 60,000 $32 \times 32$ RGB images from 100 classes. Each class contains 500 training images and 100 testing images. ImageNet-100 is composed of 100 classes with 1300 images per class for training and 500 images per class for validation. ImageNet-100 resembles real-world scenes with a higher resolution of $256 \times 256$.

**Implementation Details.** In our study, we utilized two data partitioning methods for the CIFAR-100 and ImageNet-100 datasets. Taking CIFAR-100 as an example: first method involved dividing the dataset evenly among five experts, with each expert handling 20 classes. This setup is denoted as CIFAR-100-5/20; second method involved splitting the dataset equally among ten experts, with each expert handling 10 classes. This setup is represented as CIFAR-100-10/10.

## 5.2 BASELINES SETUP AND METRICS

For all expert models, we adopt ResNet-18 as feature extractor feature. Model is optimized under Adam with learning rate $\lambda = 10^{-4}$. All methods have been evaluated using the same computation environment.

For each class within the expert, we selected 20 samples following the strategy of nearest class mean. Let $f_k(x)$ represent the feature of expert $E_k$ corresponding to input sample $x$. We calculated the class mean for the class $y$ as $\mu_y = \frac{1}{||D_y||} \sum_{x \in D_y} f_k(x)$, where $D_y$ the dataset for the class $y$. Samples were then ranked according to their $L_2$ distance to the class mean, and the top 20 samples were chosen to be included in the sample memory. We compare the top-1 average accuracy:

$$ACC = \frac{1}{N} \sum_{i=1}^{N} A_i^{acc}, \tag{14}$$

where $A_i^{acc}$ is the average accuracy of the generalist on the $i$-th expert task.

## 5.3 BENCHMARK COMPARISON

Table 1: Comparison of average accuracy.

| Method | CIFAR-100-5/20 | ImageNet-100-5/20 | CIFAR-100-10/10 | ImageNet-100-10/10 |
|---|---|---|---|---|
| KAShen (2019) | 52.0 | 55.1 | 53.0 | 55.4 |
| CFL-KALuo et al. (2019) | 55.8 | 62.0 | 57.8 | 64.6 |
| CFAde Carvalho et al. (2022) | 54.7 | 59.9 | 55.2 | 60.4 |
| DDFAXu et al. (2022) | 61.0 | 65.1 | 62.0 | 65.4 |
| Co-KA Gao et al. (2024) | 65.9 | 67.3 | 66.4 | 67.4 |
| **MEC** | **78.1** | **77.9** | **78.5** | **78.6** |

We conducted a comparison between our method and existing heterogeneous multi-teacher knowledge distillation methods, with the results summarized in Table 1. This table presents the performance metrics across two datasets (CIFAR-100 and ImageNet-100) under four different task settings, where the average accuracy for each setting is recorded. Our method consistently outperforms the baselines, showing substantial advantages across all tasks.

As shown in Table 1, traditional student models with shared output layers face challenges in heterogeneous knowledge processing, especially with an increase in the number of classes. This shared parameter structure limits the model's ability to represent information from different categories, resulting in poor alignment of heterogeneous knowledge and a classification accuracy of less than 68%.

We transition to a multi expert classifier aggregation method, where each classifier head is dedicated to handling specific classes or tasks. This design allows a single classifier head to independently process tasks and avoid conflicts. We have introduced the Helmholtz Free Energy (HFE) self normalization strategy. By using a fixed anchor to constrain the free energy of each classifier, the heterogeneous knowledge within the model is effectively aligned. These enhancements significantly

improve the scalability and stability of the model, enabling it to achieve the highest classification accuracy among all task configurations on the CIFAR-100 and ImageNet-100 datasets. Our method outperforms other heterogeneous multi teacher knowledge extraction techniques, with an average accuracy of 78%, fully verifying the effectiveness of our proposed multi-expert collaboration.

Table 2: Comparison of every expert's task accuracy.

| Dataset | Task | KA | CFL-KA | CFA | DDFA | Co-KA | MEC |
|---|---|---|---|---|---|---|---|
| CIFAR-100 | Expert1 | 52.3 | 58.0 | 57.0 | 60.9 | 65.3 | **77.1** |
| | Expert2 | 54.1 | 55.1 | 55.3 | 62.3 | 65.6 | **80.6** |
| | Expert3 | 48.2 | 56.8 | 53.2 | 60.3 | 66.8 | **76.9** |
| | Expert4 | 50.7 | 50.7 | 53.4 | 59.8 | 63.7 | **76.5** |
| | Expert5 | 54.8 | 58.6 | 54.7 | 61.9 | 68.4 | **79.2** |
| CIFAR-100 | Expert1 | 57.0 | 58.7 | 57.1 | 65.3 | 69.5 | **78.5** |
| | Expert2 | 54.5 | 59.9 | 60.2 | 60.6 | 66.1 | **79.5** |
| | Expert3 | 57.5 | 63.9 | 62.4 | 65.0 | 69.7 | **78.9** |
| | Expert4 | 48.5 | 50.7 | 48.5 | 58.5 | 61.1 | **78.2** |
| | Expert5 | 55.3 | 64.7 | 60.0 | 66.3 | 70.4 | **78.4** |
| | Expert6 | 46.5 | 50.3 | 46.7 | 60.8 | 65.8 | **77.3** |
| | Expert7 | 45.8 | 51.3 | 47.3 | 59.7 | 63.8 | **78.1** |
| | Expert8 | 56.7 | 64.0 | 65.0 | 60.7 | 69.3 | **77.3** |
| | Expert9 | 54.0 | 59.6 | 49.7 | 61.3 | 70.6 | **78.6** |
| | Expert10 | 53.8 | 55.1 | 55.1 | 61.2 | 57.5 | **80.4** |
| ImageNet-100 | Expert1 | 56.4 | 62.3 | 62.2 | 66.4 | 70.5 | **77.1** |
| | Expert2 | 55.6 | 64.1 | 57.1 | 65.6 | 64.0 | **78.0** |
| | Expert3 | 53.3 | 58.2 | 59.9 | 63.3 | 67.3 | **76.3** |
| | Expert4 | 57.5 | 60.7 | 60.2 | 67.5 | 67.6 | **79.1** |
| | Expert5 | 52.7 | 64.8 | 59.9 | 62.7 | 67.3 | **78.5** |
| ImageNet-100 | Expert1 | 49.8 | 65.5 | 62.4 | 59.8 | 71.1 | **78.8** |
| | Expert2 | 61.0 | 63.4 | 57.7 | 71.0 | 68.0 | **80.1** |
| | Expert3 | 55.4 | 71.1 | 64.0 | 65.4 | 68.7 | **77.9** |
| | Expert4 | 54.2 | 57.9 | 57.7 | 64.2 | 61.4 | **77.1** |
| | Expert5 | 54.6 | 72.3 | 60.8 | 64.6 | 70.3 | **77.7** |
| | Expert6 | 55.8 | 57.4 | 59.8 | 65.8 | 63.8 | **79.0** |
| | Expert7 | 54.8 | 55.4 | 59.6 | 64.8 | 62.4 | **78.3** |
| | Expert8 | 61.2 | 69.9 | 59.3 | 71.2 | 70.9 | **80.4** |
| | Expert9 | 48.2 | 64.8 | 65.2 | 58.2 | 71.2 | **77.1** |
| | Expert10 | 58.8 | 68.3 | 57.5 | 68.8 | 66.3 | **79.9** |

## 5.4 INDEPENDENCE AND ALIGNMENT ARE NECESSARY

We evaluated our approach of aggregating multiple expert classifiers, emphasizing the independence and alignment of heterogeneous knowledge, on the CIFAR-100-10/10 and ImageNet-100-10/10 datasets. As shown in Figure 5, increasing the number of assembled expert models leads to a corresponding growth in the number of classes to be classified. Traditional heterogeneous multi-teacher knowledge distillation methods use a single-head classifier that shares parameters across all classes, leading to interference among heterogeneous knowledge and decreased classification accuracy as the number of classes grows. In contrast, our approach employs an aggregated multi-expert classifier framework, where each classifier head specializes in different

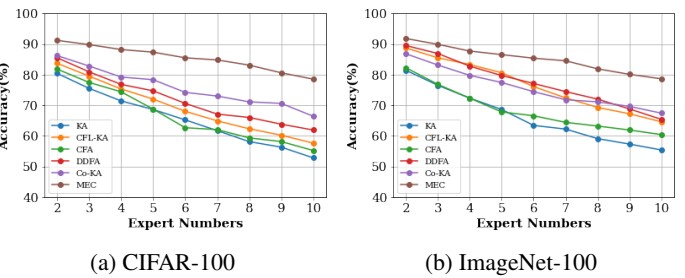

(a) CIFAR-100          (b) ImageNet-100

Figure 5: Comparison of classification accuracy of collaborate different numbers of expert models on two datasets.

classes and operates independently. This design ensures the independence and effective alignment of heterogeneous knowledge, significantly enhancing the model's scalability and generalization capabilities. Consequently, our method maintains high accuracy even as the number of expert models increases. Specifically, as shown in Table 2, across four experimental settings, our method achieves an average accuracy improvement of 10% over traditional methods for each expert task. This demonstrates that aggregating multiple expert classifiers not only allows for independent handling of different classes but also effectively aligns heterogeneous knowledge, leading to improved overall performance and accuracy.

## 5.5 ABLATION STUDY

To further verify the significance of each module in multi-expert collaboration, we conduct the ablation study as shown in Table 3.

When using only Multi-expert Representation Learning (MERL), we still adopt the single classification head for logits distillation as in heterogeneous multi-teacher knowledge distillation, but the performance appears to be relatively average. However, when Classifier Adaptation Learning (CAL) is introduced, there is a significant improvement in performance, indicating that the single head in heterogeneous multi-teacher knowledge distillation cannot

Table 3: Ablation study on CIFAR-100 and ImageNet-100.

| Dateset | MERL | CAL | Acc |
|---|---|---|---|
| | $\checkmark$ | | 61.9 |
| CIFAR-100 | | $\checkmark$ | 75.7 |
| | $\checkmark$ | $\checkmark$ | **78.5** |
| | $\checkmark$ | | 65.4 |
| ImageNet-100 | | $\checkmark$ | 75.0 |
| | $\checkmark$ | $\checkmark$ | **78.6** |

effectively handle the issue of knowledge conflict, whereas our designed CAL effectively handle the issue of knowledge conflict, whereas our designed CALeffectively addresses this problem. Finally, when combining MERL and CAL, the model achieves the best performance, demonstrating that MERL helps in building a more generalized backbone network, while CAL effectively resolves knowledge conflicts, leading to optimal overall performance.

## 6 CONCLUSION

In this paper, we addressed the critical challenge of utilizing heterogeneous teacher networks to train a student network capable of handling multiple tasks. We proposed the Multi-Expert Collaboration method, which aggregates multiple expert classifiers within the student model to ensure the independence of heterogeneous knowledge. By introducing an anchor-based HFE self-normalization strategy, we effectively aligned the knowledge from different experts, ensuring consistent energy levels across classifiers and enhancing the model's ability to integrate diverse information. Extensive experiments on the CIFAR-100 and ImageNet-100 datasets validated the superiority of our approach, showing an average accuracy improvement of over 10% compared to traditional heterogeneous multi-teacher knowledge distillation methods. The MEC framework not only improves classification accuracy but also enhances scalability and generalization capabilities. Our work provides a scalable solution for integrating diverse expert models into a unified student network, paving the way for more effective utilization of pre-trained models in multi-task learning scenarios.

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
