# OpenReview forum: "Multi-expert collaboration: Enhancing heterogeneous knowledge independence and alignment in knowledge distillation"
_ICLR.cc/2025/Conference — ICLR 2025 Conference Withdrawn Submission_

### Official Review · Reviewer_wyMP · 2024-11-02

**Soundness:** 1
**Presentation:** 2
**Contribution:** 1
**Rating:** 3
**Confidence:** 4

**Summary:**

This paper focuses on multi-teacher knowledge distillation and improves the independence between different teachers. The main idea is to apply Helmholtz Free Energy self-normalization strategy to align the features before linear classifier. The experiments on CIFAR100 and ImageNet100 present that the proposed methods outperform other multi-teacher knowledge distillation methods.

**Strengths:**

+ The prpoposed method outperforms the previous multi-teacher knowledge distillation methods.

**Weaknesses:**

+ The motivation is not clearly clarified. Why can introduction of independence between herogeneous knowledge improve the performance of multi-teacher distillation? It seems to conflict with the main idea of merging knowledge of multiple teachers. The proposed method still needs to fuse the features before classifier layer of multiple teachers.
+ This work seems meaningless. As clarified in Eq 11, the labels of all data are fully available. Why we don't train a new model from scratch, instead of training 5 models and then fusing them? I argue that the model trained by the labeled dataset can achieve similar peformance as the one obtained by this method. But training from scratch is simpler and more efficient.
+ The settings of experiment are too ideal to work on practical scenerios. The authors just split the given dataset into 5 parts to train 5 models and then merge them. I am concern about the effectiveness on real scenerios, where the given models are random.
+ Some typos, caption error about "Knowledge distillation" in abstract.

**Questions:**

+ Is the Helmholtz Free Energy in this paper just an argmax operation?

---

### Official Review · Reviewer_9HkV · 2024-11-04

**Soundness:** 2
**Presentation:** 2
**Contribution:** 2
**Rating:** 5
**Confidence:** 4

**Summary:**

The paper introduces a novel Multi-Expert Collaboration (MEC) framework to address challenges in heterogeneous multi-teacher knowledge distillation. Specifically, it proposes a multi-expert approach where each teacher’s unique knowledge contributes independently to a versatile student model, aiming to overcome issues of interference and misalignment that arise when combining knowledge from diverse sources. Inspired by Helmholtz's Free Energy (HFE) theory, MEC introduces an anchor-based self-normalization technique to harmonize knowledge from different teachers, achieving an average accuracy improvement over existing methods.

**Strengths:**

1. The paper addresses the limitations of traditional multi-teacher knowledge distillation, which often fails with heterogeneous teachers due to knowledge conflicts and misalignment.
2. The proposed approach leverages HFE theory to normalize and align the knowledge contributed by each expert classifier, which is theoretically sound and a creative application of energy-based principles.
3. The paper provides thorough experiments on CIFAR-100 and ImageNet-100 with various baseline methods, showing that MEC achieves a substantial accuracy boost.
4. The inclusion of an ablation study adds clarity to the contributions of each component (e.g., Multi-Expert Representation Learning, Classifier Adaptation Learning).

**Weaknesses:**

1. While the MEC framework shows improved accuracy, it appears computationally intensive due to multiple classifiers and additional HFE-based normalization steps. For real-time applications or resource-constrained environments, this might limit its practical utility.
2. The presentation of the MEC framework could be improved for clarity. For instance, the detailed explanation of the HFE self-normalization strategy may be challenging for some readers, as it involves advanced mathematical constructs without sufficient intuitive explanations.
3. Although the experiments cover CIFAR-100 and ImageNet-100, they are still limited to classification tasks. The MEC framework could benefit from validation across a broader range of applications, potentially highlighting its versatility or pinpointing areas where it may struggle.
4. Although MEC demonstrates significant improvements over traditional methods, additional comparison with state-of-the-art non-distillation multi-task learning approaches could provide more context on MEC’s relative benefits.

**Questions:**

1. Simplifying or visually representing key steps in the MEC pipeline could help make the approach more accessible to readers unfamiliar with energy-based methods.

2. Including a brief discussion on the computational cost or providing a comparison of training times between MEC and traditional methods would give readers insight into the efficiency of this approach.

3. To strengthen the claim that MEC effectively distills knowledge from heterogeneous sources, testing on additional types of tasks (e.g., object detection or segmentation) would be beneficial.

4. Adding more depth to the related work on energy-based methods in knowledge distillation could help situate the proposed framework within the broader context of recent advancements.

5. What is the computational overhead introduced by the MEC framework compared to traditional knowledge distillation methods?  Please provide a quantitative comparison of the training and inference times between MEC and baseline methods.

6. How does the MEC framework scale when the number of experts increases? Does performance degrade with an increasing number of expert classifiers? Are there any specific hardware or software requirements to implement MEC effectively in a real-world setting? Better to describe.

7. Please clarify or provide a step-by-step explanation of the Helmholtz Free Energy (HFE) self-normalization process and add visual representations or diagrams for complex steps in the methodology to improve accessibility for readers.

8. Is it possible to test the effectiveness of MEC on tasks beyond classification, such as object detection or segmentation?

9. How does MEC compare to non-distillation multi-task learning methods? Is it possible to include these comparisons to highlight MEC's advantages more comprehensively?

10. Please provide quantitative comparisons of computational costs or runtime between MEC and baseline methods. This would allow for a more concrete assessment of the practical tradeoffs.

---

### Official Review · Reviewer_t7Rg · 2024-11-04

**Soundness:** 3
**Presentation:** 3
**Contribution:** 2
**Rating:** 5
**Confidence:** 4

**Summary:**

The paper introduces a novel method called Multi-Expert Collaboration (MEC) for addressing the challenges of utilizing heterogeneous teacher networks to train a versatile student model capable of handling multiple tasks. The proposed method aggregates multiple expert classifiers within the student model to ensure the independence of heterogeneous knowledge. Additionally, it employs an anchor-based Helmholtz Free Energy (HFE) self-normalization strategy to align the knowledge from different experts, ensuring consistent energy levels across classifiers.

**Strengths:**

Comprehensive Approach: The paper proposes a multi-expert collaboration method that addresses both the independence and alignment issues of heterogeneous knowledge in knowledge distillation. The extensive experiments on CIFAR-100 and ImageNet-100 datasets provide strong empirical evidence of the method's effectiveness.

**Weaknesses:**

1. Limited Innovation: The paper primarily relies on existing theories and techniques to address the problem. The paper lacks significant theoretical breakthroughs or methodological innovations that could set it apart from existing literature.
2. Comparative Analysis:The paper does not provide a thorough comparative analysis with current state-of-the-art methods. It would be beneficial to include comparisons with more recent and advanced methods in the field to provide a more comprehensive evaluation of the proposed approach.
3. Generalizability:The experiments are conducted on CIFAR-100 and ImageNet-100 datasets, which are relatively small and well-studied. It would be valuable to test the proposed method on larger and more diverse datasets to evaluate its generalizability and robustness.

**Questions:**

see weaknesses

---

### Official Review · Reviewer_ZSY5 · 2024-11-04

**Soundness:** 3
**Presentation:** 2
**Contribution:** 2
**Rating:** 3
**Confidence:** 4

**Summary:**

The paper highlights an issue of mututal inference betweeen heterogenous knowledge in th existing multi-teacher knowledge distillation setup. It points out merely concatenating logits from different teachers for the case when the data they are trained on is sufficiently different, leads to suboptimal performance. Instead they propose a multi-expert collaborative approach for this setup by modifying this distillation at an architectural level by decoupling feature representation learning and classifer adaptation. Here,  they first construct a universal feature extraction backbone by projecting expert features and student (generalist) features into a common space and then use two loss functions: first one to align the features of the student with the experts in the common space and second one to learn appropriate projection by reconstructing the initial features of each of the experts. For the alignment they just the pairwise KL loss between the student and each of the teachers and feature reconstruction is just standard L-2 loss. Furthermore, for the classification part, they now use this learned feature representation of the student and propose an adpation strategy where they start the student classifier from the weights of expert classifiers (as the number of heads are same) and then use the concept of Helmholtz free energy (HFE) being higher for in-dsitribution data for a particular expert. They also propose an additional constraint to align the HFE and the KD training along with the CE loss.
Then, for prediction they select the expert classifier with the highest HFE. To validate their hypothesis, they perform experiments on CIFAR-100, ImageNet-100 datasets with two different setups of 5 and 10 experts showing significant gains. Furthermore, they also perform an ablation study to verify the significance of each of their modules.

**Strengths:**

The issue highllighted and the problem tackled in the paper are quite important given the lately trending foundational models era. Furthermore, the proposed scheme including the coommon feature space representation learning using the projection and then adapting the student backbone and the mutli-headed classifier initialized from experts followed by using helmholtz free energy for prediction seems sound and intuitive. On top of this, significant gains in the performance over latest methods involving knowledge amalgamation sounds promising. Furthermore, the ablation study provided for advocating the importance of each of the modules is also useful.

**Weaknesses:**

Even though as said in the strengths section method is intuitive still this idea of shared representation learning and decoupled distillation also exists in previous papers like the Co-KA paper mentioned and also used as a baseline (upon taking a quick look at that paper the modules have similar functionality although being technically different). Dissecting the technical contribution by analyzing each module, there is not much of a novelty factorm in the multi-expert representation learning since both KL scattering and projection to some common feature space seems pretty obvious in setups similar to this like domain adaptation which also involves learning domain invariant features. For the second module I do understand they propose the additional HFE constraint but the KD loss is pretty standard which further brings a question that is this constraint responsible (with the joint training) that is giving such gains or it is the modified inference setup?
Even though jointly combining th modules is resulting in significant gains still exact novel contribution of the paper (although is there) might be in between limited to significant, with more closer to limited.

On the experimental part, even though as said in the strengths section that the numbers look promising still the experiments shown in the paper are not-at-all enoguh to verify the use case of this paper in this large scale era. They haven't shown even some multi-task setup having different datasets or varied domain setup, they have just shown a seggregation based on classes setup. There should be sufficient experiments involving data collected from separate sources of same/different classes each expert handling a different amount of data analysis, some overlapping classes and more use-cases inspired from real-world possibilites. Furthermore, the scale of datasets used is very small. Usually the gains might diminish in larger scales when sufficient data is available with each expert or some classes (may be 5-10 \%) might overlap. Secondly the architecture used is ResNet-18 which again makes it difficult to validate the approach and advocate for its use case at a larger scale. Lately most of the works evolve around atleast ResNet-50 or latest ViT architectures or even the large scae pretrained ones like CLIP, LiT etc.

**Questions:**

Please see the concerns raised in the weakness section. Initially I am going with the reject but I would be happy to reconsider based on two justifications: (a) Primarliy on the lack of sufficient empirical validity as highlighted in the weakness section. (b) Secondarily a question that even though the structure/layout of the approach is similar to Co-KA method and the multi-expert representation learning is pretty obvious from the literature and the adaption module has some contribution: 1. What is the overall extra technical contribution of this paper (please highlight clearly in the paper also) as againt all of the previous methods? 2. Further intuition/analysis of how this contribution on top of the previous methods is actually leading to such huge gains (even though the experimental setup is not quite realistic) like is it due to the modified multiheaded sceheme/HFE constraint or due to just the modified inference?

---

### Note · Authors · 2024-11-26

I have read and agree with the venue's withdrawal policy on behalf of myself and my co-authors.